# OpenReview forum: "Emergence of grid-like representations by training recurrent neural networks to perform spatial localization"
_ICLR.cc/2018/Conference — Accept (Poster)_

### Official Review · AnonReviewer1 · 2017-11-22
**Interesting paper, and quite thorough work**

**Rating:** 8
**Confidence:** 4

**Review:**

The authors train an RNN to perform deduced reckoning (ded reckoning) for spatial navigation, and then study the responses of the model neurons in the RNN. They find many properties reminiscent of neurons in the mammalian entorhinal cortex (EC): grid cells, border cells, etc. When regularization of the network is not used during training, the trained RNNs no longer resemble the EC. This suggests that those constraints (lower overall connectivity strengths, and lower metabolic costs) might play a role in the EC's navigation function.

The paper is overall quite interesting and the study is pretty thorough: no major cons come to mind. Some suggestions / criticisms are given below.

1) The findings seem conceptually similar to the older sparse coding ideas from the visual cortex. That connection might be worth discussing because removing the regularizing (i.e., metabolic cost) constraint from your RNNS makes them learn representations that differ from the ones seen in EC. The sparse coding models see something similar: without sparsity constraints, the image representations do not resemble those seen in V1, but with sparsity, the learned representations match V1 quite well. That the same observation is made in such disparate brain areas (V1, EC) suggests that sparsity / efficiency might be quite universal constraints on the neural code.

2) The finding that regularizing the RNN makes it more closely match the neural code is also foreshadowed somewhat by the 2015 Nature Neuro paper by Susillo et al. That could be worthy of some (brief) discussion.

Sussillo, D., Churchland, M. M., Kaufman, M. T., & Shenoy, K. V. (2015). A neural network that finds a naturalistic solution for the production of muscle activity. Nature neuroscience, 18(7), 1025-1033.

3) Why the different initializations for the recurrent weights for the hexagonal vs other environments? I'm guessing it's because the RNNs don't "work" in all environments with the same initialization (i.e., they either don't look like EC, or they don't obtain small errors in the navigation task). That seems important to explain more thoroughly than is done in the current text.

4) What happens with ongoing training? Animals presumably continue to learn throughout their lives. With on-going (continous) training, do the RNN neurons' spatial tuning remain stable, or do they continue to "drift" (so that border cells turn into grid cells turn into irregular cells, or some such)? That result could make some predictions for experiment, that would be testable with chronic methods (like Ca2+ imaging) that can record from the same neurons over multiple experimental sessions.

5) It would be nice to more quantitatively map out the relation between speed tuning, direction tuning, and spatial tuning (illustrated in Fig. 3). Specifically, I would quantify the cells' direction tuning using the circular variance methods that people use for studying retinal direction selective neurons. And I would quantify speed tuning via something like the slope of the firing rate vs speed curves. And quantify spatial tuning somehow (a natural method would be to use the sparsity measures sometimes applied to neural data to quantify how selective the spatial profile is to one or a few specific locations). Then make scatter plots of these quantities against each other. Basically, I'd love to see the trends for how these types of tuning relate to each other over the whole populations: those trends could then be tested against experimental data (possibly in a future study).

---

> ### Author Response · Authors · 2018-01-05
> **reply to reviewer 1**
>
> Thank you for your positive assessment and feedback.
>
> The reviewer raised the interesting point on the potential connection between the sparse coding work by Olshausen and Field (1996) and our work (note that Reviewer 3 also raised the same point). Indeed, the initial idea of this work is partly inspired by sparse coding. In the Discussion section, we have now included a detailed discussion of the relation of our work to the sparse coding work. While there are important conceptual similarities, there are also some important differences that need to be mentioned. For example, the grid-like response patterns are averaged responses, and are not linear filters based on the input to the network as the Gabor filters. In our network, the velocity response needs to be integrated and maintained in order for the neurons to be spatially tuned. Also, the sparsity constraint in Olshausen and Field could be interpreted as imposing a particular form of Bayesian prior, while in our work, we find it difficult to interpret that way. We also include a discussion on Sussillo et al. (2015), which we agree is very relevant to our work. Thank you for both suggestions.
>
> Concerning the different initializations, we want to clarify that we tried a few different initializations and found the results to be robust. As a way to show the robustness, we have presented the results with different initializations. We have slightly changed the text to make this point more explicit. We apologize for the confusion.
>
> On the question of ongoing training- we think this is a great idea. We have thought about these issues, including training the same network to perform tasks in various environments, but on the other hand, we also think a systematic treatment of these issues goes beyond the scope of the current paper, which is just a proof-of-principle of the approach.
>
> The reviewer also suggests a more quantitative characterization of the relation between different kinds of tuning properties. We have computed the selectivity indexes for different tuning properties along these lines. It appears that at the population level, the relation between these different indexes is complex. We now incorporate a figure in the Appendix. We also examined the experimental literature on the relation between different types of tuning in EC, and there is not enough evidence to draw any conclusions. Thus, we think this is an interesting and important question that should be informative for future experiments. But one might need some more sophisticated ways to perform this analysis in order to better reveal the dependence between different types of tuning. For example, maybe one could first do a systematic clustering based on the neurons’ response, and examine the dependence of the tuning based on the clustering. We would like to look deeper into these issues in the near future. Thank you for this suggestion.

---

### Official Review · AnonReviewer3 · 2017-11-25
**Great paper on emergence of representations as observed in mammals**

**Rating:** 9
**Confidence:** 4

**Review:**

Congratulations on a very interesting and clear paper.  While ICLR is not focused on neuroscientific studies, this paper clearly belongs here as it shows what representations develop in recurrent networks that are trained on spatial navigation. Interestingly, these include representations that have been observed in mammals and that have attracted considerable attention, even honored with a Nobel prize.

I found it is very interesting that the emergence of these representations was contingent on some regularization constraint. This seems similar to the visual domain where edge detectors emerge easily when trained on natural images with sparseness constraints as in Olshausen&Field and later reproduced with many other models that incorporate sparseness constraints.

I do have some questions about the training itself. The paper mentions a metabolic cost that is not specified in the paper. This should be added.

My biggest concern is about Figure 6a. I am puzzled why is the error is coming down before the boundary interaction? Even more puzzling, why does this error go up again for the blue curve (no interaction)? Shouldn’t at least this curve be smooth?

---

> ### Author Response · Authors · 2018-01-05
> **reply to reviewer 3**
>
> Thank you for your positive assessment and feedback.
>
> The reviewer raised the interesting point on the potential connection between the sparse coding work by Olshausen and Field (1996) and our work (note that Reviewer 1 raised a similar point). Indeed, the initial idea behind this work is partly inspired by sparse coding. In the Discussion section, we have now included a detailed discussion of the relation of our work to the sparse coding work. We also think while there are important conceptual similarities, there are also some important differences that need to be mentioned. For example, the grid-like response patterns are averaged responses, they are not linear filters based on the input to the network as the Gabor filters. In our network, the velocity response needs to be integrated and maintained in order for the neurons to be spatially tuned. Also, the sparsity constraint in Olshausen and Field could be interpreted as imposing a particular form of Bayesian prior, while in our work, we find it difficult to interpret that way. We also add discussions on a few other related studies. We thank the reviewer for this suggestion.
>
> In terms of Figure 6a, the increase of the blue curve after the boundary interaction is due to the accumulation of noise in the RNN, which would gradually increase the error (roughly linearly, thinking about Brownian motion may help here) without further boundary interactions. We now explain this in the caption of Figure 6. The reason the blue curve is not entirely smooth is that it is the averaged error of a set of sample paths, and the number of paths is not very large. However, we think the general pattern is clear. In terms of why the error is coming down before the boundary interaction, one possible yet speculative reason is that the boundary-related firing has a certain width, thus the error-correction might already start before the boundary interaction. In fact, this would be a natural interpretation based on a previous model which used boundary responses to correct the grid-drift (Hardcastle et al., Neuron, 2015). We note that one caveat to that interpretation is that, in our simulations, sometimes we don’t observe such a drop in error before the boundary interactions. Also, it remains unclear if the error-correction mechanisms in our model are the same as those proposed previously. However, the consistent pattern we observe empirically is that error robustly reduced after the boundary interaction. We thus have focused on this robust empirical observation, without speculating on an exact mechanism, which remains to be elucidated in the future.

---

### Official Review · AnonReviewer2 · 2017-11-27
**An interesting paper**

**Rating:** 8
**Confidence:** 4

**Review:**

This paper aims at better understanding the functional role of grid cells found in the entorhinal cortex by training an RNN to perform a navigation task.

On the positive side:

This is the first paper to my knowledge that has shown that grid cells arise as a product of a navigation task demand. I enjoyed reading the paper which is in general clearly written. I have a few, mostly cosmetic, complaints but this can easily be addressed in a revision.

On the negative side:

The manuscript is not written in a way that is suitable for the target ICLR audience which will include, for the most part, readers that are not expert on the entorhinal cortex and/or spatial navigation.

First, the contributions need to be more clearly spelled out. In particular, the authors tend to take shortcuts for some of their statements. For instance, in the introduction, it is stated that previous attractor network type of models (which are also recurrent networks) “[...] require hand-crafted and fined tuned connectivity patterns, and the evidence of such specific 2D connectivity patterns has been largely absent.” This statement is problematic for two reasons:

(i) It is rather standard in the field of computational neuroscience to start from reasonable assumptions regarding patterns of neural connectivity then proceed to show that the resulting network behaves in a sensible way and reproduces neuroscience data. This is not to say that demonstrating that these patterns can arise as a byproduct is not important, on the contrary. These are just two complementary lines of work. In the same vein, it would be silly to dismiss the present work simply because it lacks spikes.

(ii) the authors do not seem to address one of the main criticisms they make about previous work and in particular "[a lack of evidence] of such specific 2D connectivity patterns". My understanding is that one of the main assumptions made in previous work is that of a center-surround pattern of lateral connectivity. I would argue that there is a lot of evidence for local inhibitory connection in the cortex. Somewhat related to this point, it would be insightful to show the pattern of local connections learned in the RNN to see how it differs from the aforementioned pattern of connectivity.

Second, the navigation task used needs to be better justified. Why training a network to predict 2D spatial location from velocity inputs? Why is this a reasonable starting point to study the emergence of grid cells? It might be obvious to the authors but it will not be to the ICLR audience. Dead-reckoning (i.e., spatial localization from velocity inputs) is of critical ecological relevance for many animals. This needs to be spelled out and a reference needs to be added.  As a side note, I would have expected the authors to use actual behavioral data but instead, the network is trained using artificial trajectories based on "modified Brownian motion”. This seems like an important assumption of the manuscript but the issue is brushed off and not discussed. Why is this a reasonable assumption to make? Is there any reference demonstrating that rodent locomotory behavior in a 2D arena is random?

Figure 4 seems kind of strange. I do not understand how the “representative units” are selected and where the “late” selectivity on the far right side in panel a arises if not from “early” units that would have to travel “far” from the left side… Apologies if I am missing something obvious.

I found the study of the effect of regularization to be potentially the most informative for neuroscience but it is only superficially treated. It would have been nice to see a more systematic treatment of the specifics of the regularization needed to get grid cells.

---

> ### Author Response · Authors · 2018-01-05
> **reply to reviewer 2**
>
> Thank you for your positive assessment and feedback. Below we address the concerns raised in the review. We have taken the suggestions in the review to make this manuscript as accessible to the general ICLR audience as possible.
>
> 1. Concerning the attractor model to motivate our model, we agree with the reviewer that our initially statement is unclear and potentially misleading. We have thus revised the statements in the Introduction to better motivate our work. Specifically, we now present a more balance view of the two types of models, and explicitly present our modeling as a complementary/alternative approach. Also we are now more specific about the assumptions made in the attractor network model. In particular we point out that it is the subtle asymmetry in the weights we are mostly concerned about, not the local inhibition. We agree with the reviewer that local inhibition is not uncommon in cortex. However, these models also assume systematic weight asymmetry in the connectivity matrix, which is responsible for the movement of the bump attractor for tracking the animal’s location. Without such asymmetry, grids would not emerge in these models. For example, please see Eq. (2) in Couey et al. "Recurrent inhibitory circuitry as a mechanism for grid formation." Just to be clear, we are not claiming that such connectivity pattern does not exist in the brain. We are just making the points that 1) this assumed pattern seems too restrictive, 2) so far there is no evidence for it. Furthermore, now we also motivate our work by the need for having a model that could account for other types of response patterns in Entorhinal Cortex. Note that such issues have been largely ignored previously.
> As for the connections in our trained RNN, we did perform some preliminary analysis. We now include a discussion on this point in the final paragraph of the Discussion.
>
> 2. Concerning the justification of the navigation task-
> We completely agree that it would be useful to make the ecological relevance of dead-reckoning task explicit, in particular for the ICLR audience. We have discussed this point and add relevant references there. We thank the reviewer for this helpful suggestion.
>
> On the issues of using artificial v.s. animal’s real trajectory to train the model- Historically there is a long debate in neuroscience on the use of artificial or naturalistic stimuli, in particular for vision. For example, see Rust and Movshon "In praise of artifice." In the present paper, we didn’t use the animal’s real running trajectories for multiple reasons. First, we didn’t have access to enough data of animal’s running trajectories. Second, from a theoretical point of view, it seems advantageous to start with simple trajectories that we have complete control over. Third, grid-like responses have been reported in various animal species, including rat, mouse, bat and humans. The locomotory behavior of these different animal species can be quite different. For example, there are qualitative differences between rats and mice. We thus started from simple movement statistics to see what we can get. Having said that, we do agree with the reviewer that it would be very interesting to test the model using real trajectories. We briefly discussed these points in section 2.2. Also, as a side note, we have been talking to experimental colleagues to get data of the mice’s running behavior.
>
> 3. The representative units are picked manually based on their spatial firing pattern. We simply identified a few units that clearly show grid-like or boundary-related responses, and marked them after the dimensionality reduction using t-SNE. We want to emphasize that the main point of Figure 4 is to show that early during training all responses are similar to the boundary related responses, and only as training continues do the grid-like units emerge. This did not have to be true. Early in training some units could have boundary responses and others could have grid-like responses. If these spatial response patterns remained largely unchanged during training then all units would travel a short distance in the t-SNE figure. This is in fact the case for the boundary responses: the final spatial responses are very close to the responses that developed early in training and so these units do not travel much in this space. We have modified the text to make it more clear. Thank you for raising this point.
>
> 4. In terms of the role of regularization, we agree that it is potentially interesting from the neuroscience point of view. While we think the most informative effect remains the one we show in the main text, we now also include a new figure in the Appendix for the triangular environment to better illustrate the effect of the metabolic cost and adding noise to the RNN units (without being too redundant to the figure in the main text). We would be happy to include more simulations on this if the reviewer thinks that’s necessary.

---

### Author Response · Authors · 2018-01-05
**reply to all reviewers**

We thank all three reviewers for their careful reading and the positive assessment of our manuscript. We appreciate the reviewers’ detailed and constructive feedback, which have helped us substantially improve various aspects of the manuscript, in particular the presentation of our results. We have revised and uploaded a new version of the manuscript, taking into account the reviewers’ suggestions.

---

### Decision · Program_Chairs · 2018-01-29
**ICLR 2018 Conference Acceptance Decision**

**Decision:**

Accept (Poster)

**Comment:**

This work shows how activation patterns of units reminiscent of grid and border cells emerge in RNNs trained on navigation tasks. While the ICLR audience is not mainly focused on neuroscience, the findings of the paper are quite intriguing, and grid cells are sufficiently well-known and "mainstream" that this may interest many people.